# Geometry-complete diffusion for 3D molecule generation and optimization
Alex Morehead ⓘ ✉ & Jianlin Cheng ⓘ

Generative deep learning methods have recently been proposed for generating 3D molecules using equivariant graph neural networks (GNNs) within a denoising diffusion framework. However, such methods are unable to learn important geometric properties of 3D molecules, as they adopt molecule-agnostic and non-geometric GNNs as their 3D graph denoising networks, which notably hinders their ability to generate valid large 3D molecules. In this work, we address these gaps by introducing the Geometry-Complete Diffusion Model (GCDM) for 3D molecule generation, which outperforms existing 3D molecular diffusion models by significant margins across conditional and unconditional settings for the QM9 dataset and the larger GEOM-Drugs dataset, respectively. Importantly, we demonstrate that GCDM's generative denoising process enables the model to generate a significant proportion of valid and energetically-stable large molecules at the scale of GEOM-Drugs, whereas previous methods fail to do so with the features they learn. Additionally, we show that extensions of GCDM can not only effectively design 3D molecules for specific protein pockets but can be repurposed to consistently optimize the geometry and chemical composition of existing 3D molecules for molecular stability and property specificity, demonstrating new versatility of molecular diffusion models. Code and data are freely available on GitHub.

Generative modeling has recently been experiencing a renaissance in modeling efforts driven largely by denoising diffusion probabilistic models (DDPMs). At a high level, DDPMs are trained by learning how to denoise a noisy version of an input example. For example, in the context of computer vision, Gaussian noise may be successively added to an input image with the goals of a DDPM in mind. We would then desire for a generative model of images to learn how to successfully distinguish between the original input image's feature signal and the noise added to the image thereafter. If a model can achieve such outcomes, we can use the model to generate new images by first sampling multivariate Gaussian noise and then iteratively removing, from the current state of the image, the noise predicted by the model. This classic formulation of DDPMs has achieved significant results in the space of image generation[1], audio synthesis[2], and even meta-learning by learning how to conditionally generate neural network checkpoints[3]. Furthermore, such an approach to generative modeling has expanded its reach to encompass scientific disciplines such as computational biology[4–8], computational chemistry[9–11], and computational physics[12].

Concurrently, the field of geometric deep learning[13] has seen a sizeable increase in research interest lately, driven largely by theoretical advances within the discipline[14] as well as by applications of such methodology[15–18]. Notably, such applications even include what is considered by many

researchers to be a solution to the problem of predicting 3D protein structures from their corresponding amino acid sequences[19]. Such an outcome arose, in part, from recent advances in sequence-based language modeling efforts[20,21] as well as from innovations in equivariant neural network modeling[22].

However, it is currently unclear how the expressiveness of geometric neural networks impacts the ability of generative methods that incorporate them to faithfully model a geometric data distribution. In addition, it is currently unknown whether diffusion models for 3D molecules can be repurposed for important, real-world tasks without retraining or fine-tuning and whether geometric diffusion models are better equipped for such tasks. Toward this end, in this work, we provide the following findings:

- Neural networks that perform message-passing with geometric quantities enable diffusion generative models of 3D molecules to generate valid and energetically-stable large molecules whereas non-geometric message-passing networks fail to do so, where we introduce key computational metrics to enable such findings.
- Physical inductive biases such as invariant graph attention and molecular chirality both play important roles in diffusion-generating valid 3D molecules.

Department of Electrical Engineering & Computer Science, NextGen Precision Health, University of Missouri, Columbia, MO 65211, USA.
✉e-mail: acmwhb@missouri.edu

- Our newly-proposed Geometry-Complete Diffusion Model (GCDM —see Fig. 1), which is the first diffusion model to incorporate the above insights and achieve the ideal type of equivariance for 3D molecule generation (i.e., SE(3) equivariance), establishes state-of-the-art (SOTA) results for conditional 3D molecule generation on the QM9 dataset as well as for unconditional molecule generation on the GEOM-Drugs dataset of large 3D molecules, for the latter more than doubling PoseBusters validity rates; generates more unique and novel small molecules for unconditional generation on the QM9 dataset; and achieves better Vina energy scores and more than twofold higher PoseBusters validity rates[23] for protein-conditioned 3D molecule generation.

- We further demonstrate that geometric diffusion models such as GCDM can consistently perform 3D molecule optimization for molecular stability as well as for specific molecular properties without requiring any retraining and can consistently do so whereas non-geometric diffusion models cannot.

## Results and discussion

### Unconditional 3D molecule generation—QM9

The first dataset used in our experiments, the QM9 dataset[24], contains molecular properties and 3D atom coordinates for 130k small molecules. Each molecule in QM9 can contain up to 29 atoms after hydrogen atoms are imputed for each molecule following dataset postprocessing as in ref. 25. For the task of 3D molecule generation, we train GCDM to unconditionally generate molecules by producing atom types (H, C, N, O, and F), integer atom charges, and 3D coordinates for each of the molecules' atoms. Following ref. 26, we split QM9 into training, validation, and test partitions consisting of 100k, 18k, and 13k molecule examples, respectively.

**Metrics.** We measure each method's average negative log-likelihood (NLL) over the corresponding test dataset, for methods that report this quantity. Intuitively, a method achieving a lower test NLL compared to other methods indicates that the method can more accurately predict

denoised pairings of atom types and coordinates for unseen data, implying that it has fit the underlying data distribution more precisely than other methods. In terms of molecule-specific metrics, we adopt the scoring conventions of ref. 27 by using the distance between atom pairs and their respective atom types to predict bond types (single, double, triple, or none) for all but one baseline method (i.e., E-NF). Subsequently, we measure the proportion of generated atoms that have the right valency (atom stability—AS) and the proportion of generated molecules for which all atoms are stable (molecule stability—MS). To offer additional insights into each method's behavior for 3D molecule generation, we also report the validity (Val) of the generated molecules as determined by RDKit[28], the uniqueness of the generated molecules overall (Uniq), and whether the generated molecules pass each of the de novo chemical and structural validity tests (i.e., sanitizable, all atoms connected, valid bond lengths and angles, no internal steric clashes, flat aromatic rings and double bonds, low internal energy, correct valence, and kekulizable) proposed in the PoseBusters software suite[23] and adopted by recent works on molecule generation tasks[29,30]. Each method's results in the top half (bottom half) of Table 1 are reported as the mean and standard deviation (mean and Student's t-distribution 95% confidence error intervals) (±) of each metric across three (five) test runs on QM9, respectively.

**Baselines.** Besides including a reference point for molecule quality metrics using QM9 itself (i.e., Data), we compare GCDM (a geometry-complete DDPM - i.e., GC-DDPM) to 10 baseline models for 3D molecule generation, each trained and tested using the same corresponding QM9 splits for fair comparisons: G-Schnet[31]; Equivariant Normalizing Flows (E-NF)[27]; Graph Diffusion Models (GDM)[25] and their variations (i.e., GCM-aug); Equivariant Diffusion Models (EDM)[25]; Bridge and Bridge + Force[32]; latent diffusion models (LDMs) such as GraphLDM and its variation GraphLDM-aug[33]; as well as the state-of-the-art GeoLDM method[33]. Note that we specifically include these baselines as representative implicit bond prediction methods for which bonds are inferred using their generated molecules' atom types and inter-

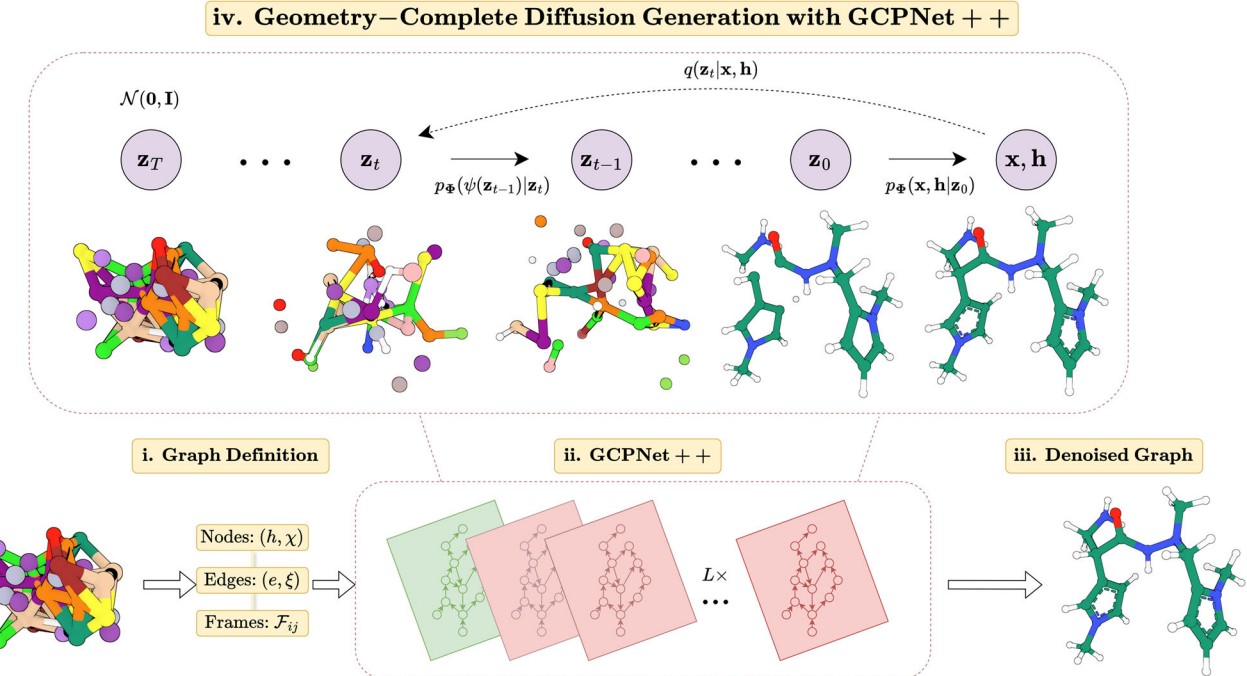

**Fig. 1 | A framework overview of the proposed geometry-complete diffusion model (GCDM) for geometric and chirality-aware 3D molecule generation.** The framework consists of (i) a graph (topology) definition process; (ii) a GCPNET-based graph neural network for SE(3)-equivariant graph representation learning; (iii) denoising of 3D input graphs using GCPNET++; and (iv) application of a trained GCPNET++ denoising network for 3D molecule generation. Zoom in for the best viewing experience.

## Table 1 | Comparison of GCDM with baseline methods for 3D molecule generation

| Type | Method | NLL ↓ | AS (%) ↑ | MS (%) ↑ | Val (%) ↑ | Val and Uniq (%) ↑ | | |
|---|---|---|---|---|---|---|---|---|
| NF | E-NF | −59.7 | 85.0 | 4.9 | 40.2 | 39.4 | | |
| Generative GNN | G-Schnet | – | 95.7 | 68.1 | 85.5 | 80.3 | | |
| DDPM | GDM | −94.7 | 97.0 | 63.2 | – | – | | |
| | GDM-aug | −92.5 | 97.6 | 71.6 | 90.4 | 89.5 | | |
| | EDM | −110.7 ± 1.5 | 98.7 ± 0.1 | 82.0 ± 0.4 | 91.9 ± 0.5 | 90.7 ± 0.6 | | |
| | Bridge | – | 98.7 ± 0.1 | 81.8 ± 0.2 | – | 90.2 | | |
| | Bridge + Force | – | 98.8 ± 0.1 | 84.6 ± 0.3 | 92.0 | 90.7 | | |
| LDM | GraphLDM | – | 97.2 | 70.5 | 83.6 | 82.7 | | |
| | GraphLDM-aug | – | 97.9 | 78.7 | 90.5 | 89.5 | | |
| | GeoLDM | – | **98.9** ± 0.1 | **89.4** ± 0.5 | 93.8 ± 0.4 | 92.7 ± 0.5 | | |
| GC-DDPM—Ours | GCDM w/o Frames | <u>−162.3</u> ± 0.3 | 98.4 ± 0.0 | 81.7 ± 0.5 | <u>93.9</u> ± 0.1 | <u>92.7</u> ± 0.1 | | |
| | GCDM w/o SMA | −131.3 ± 0.8 | 95.7 ± 0.1 | 51.7 ± 1.4 | 83.1 ± 1.7 | 82.8 ± 1.7 | | |
| | GCDM | **−171.0** ± 0.2 | <u>98.7</u> ± 0.0 | <u>85.7</u> ± 0.4 | **94.8** ± 0.2 | **93.3** ± 0.0 | | |
| Data | | | 99.0 | 95.2 | 97.7 | 97.7 | | |

| Method | NLL ↓ | AS (%) ↑ | MS (%) ↑ | Val (%) ↑ | Val and Uniq (%) ↑ | Novel (%) ↑ | PB-Valid (%) ↑ |
|---|---|---|---|---|---|---|---|
| GeoLDM | – | **98.9** ± 0.0 | **89.8** ± 0.4 | <u>93.6</u> ± 0.2 | <u>91.8</u> ± 0.2 | <u>53.5</u> ± 0.6 | **93.1** ± 0.4 |
| GCDM | **−169.4** ± 0.8 | <u>98.7</u> ± 0.1 | <u>86.0</u> ± 0.7 | **94.9** ± 0.3 | **93.4** ± 0.3 | **58.7** ± 0.5 | 91.9 ± 0.5 |

The results in the top half of the table are reported in terms of the negative log-likelihood (NLL) $-\log p(\mathbf{x}, \mathbf{h}, N)$, atom stability, molecule stability, validity, and uniqueness of 10,000 samples drawn from each model, with standard deviations (±) for each model across three runs on QM9. The results in the bottom half of the table are for methods specifically evaluated across five runs on QM9 using Student's $t$-distribution 95% confidence intervals for per-metric errors, additionally with novelty (Novel) defined as the percentage of (valid and unique) generated molecule SMILES strings that were not found in the QM9 dataset and PoseBusters validity (PB-Valid) defined as the percentage of generated molecules that pass all relevant de novo structural and chemical sanity checks listed in the "Unconditional 3D molecule generation—QM9" section. The top-1 (best) results for this task are in bold, and the second-best results are underlined, with—denoting a metric value that is not available.

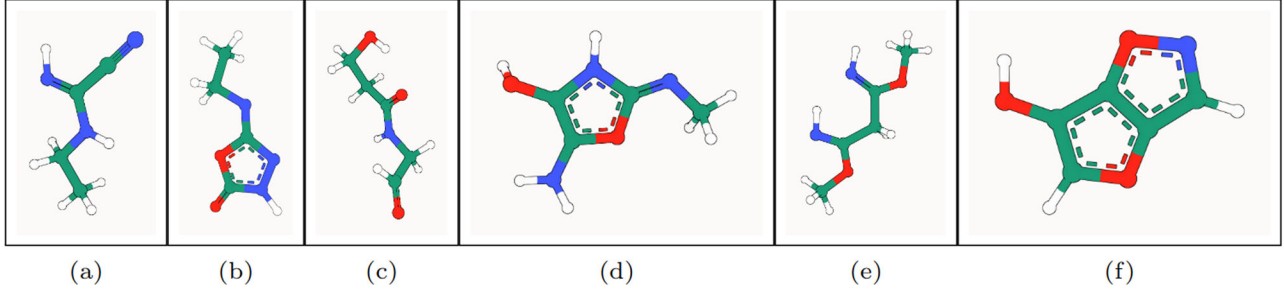

(a) (b) (c) (d) (e) (f)

**Fig. 2 | PB-valid 3D molecules generated by GCDM for the QM9 dataset.** The corresponding SMILES strings for these generated small molecules, from left to right, are as follows: **a** [H]/N=C(\C#N)NCC, **b** CC[N]c1n[nH]c(=O)o1, **c** O=CCNC(=O)CCO, **d** C/N=c1/[nH]c(O)c(N)o1, **e** [H]/N=C(/C[C]([NH])OC)OC, and **f** Oc1coc2cnoc12.

atom distances, in contrast to explicit bond prediction approaches such as those of refs. 34,35 for fair comparisons with our method. For each of such baseline methods, we report their results as curated by refs. 32,33. We further include two GCDM ablation models to more closely analyze the impact of certain key model components within GCDM. These two ablation models include GCDM without chiral and geometry-complete local frames $\mathcal{F}_{ij}$ (i.e., GCDM w/o Frames) and GCDM without scalar message attention (SMA) applied to each edge message (i.e., GCDM w/o SMA). In "Methods" section as well as Supplementary Methods A.2 and Supplementary Note B, we further discuss GCDM's design, hyperparameters, and optimization with these model configurations.

**Results**. In the top half of Table 1, we see that GCDM achieves the highest percentage of probable (NLL), valid, and unique molecules compared to all baseline methods, with AS and MS results marginally lower than those of GeoLDM yet with lower standard deviations. In the bottom half of Table 1, where we reevaluate GCDM and GeoLDM using 5 sampling runs and report 95% confidence intervals for each metric, GCDM generates 1.6% more RDKit-valid and unique molecules and 5.2% more novel molecules compared to GeoLDM, all while offering the

best reported NLL for the QM9 test dataset. This result indicates that although GeoLDM offers novelty rates close to parity (i.e., 50%), GCDM nearly matches the stability and PB-validity rates of GeoLDM while yielding novel molecules nearly 60% of the time on average, suggesting that GCDM may prove more useful for accurately exploring the space of novel yet valid small molecules. Our ablation of SMA within GCDM demonstrates that, to generate stable 3D molecules, GCDM heavily relies on both being able to perform a lightweight version of fully-connected graph self-attention[20], which suggests avenues of future research that will be required to scale up such generative models to large biomolecules such as proteins. Additionally, removing geometric local frame embeddings from GCDM reveals that the inductive biases of molecular chirality and geometry-completeness are important contributing factors in GCDM achieving these SOTA results. Figure 2 illustrates PoseBusters-valid examples of QM9-sized molecules generated by GCDM.

### Property-conditional 3D molecule generation—QM9
**Baselines**. Towards the practical use case of conditional generation of 3D molecules, we compare GCDM to existing E(3)-equivariant models, EDM[25] and GeoLDM[33], as well as to two naive baselines: "Naive (Upper-

**Table 2 | Comparison of GCDM with baseline methods for property-conditional 3D molecule generation**

| Task | $\alpha\downarrow$ | $\Delta\epsilon\downarrow$ | $\epsilon_{HOMO}\downarrow$ | $\epsilon_{LUMO}\downarrow$ | $\mu\downarrow$ | $C_v\downarrow$ |
|---|---|---|---|---|---|---|
| Units | Bohr$^3$ | meV | meV | meV | D | $\frac{cal}{mol}$ K |
| Naive (Upper-bound) | 9.01 | 1470 | 645 | 1457 | 1.616 | 6.857 |
| # Atoms | 3.86 | 866 | 426 | 813 | 1.053 | 1.971 |
| EDM | 2.76 | 655 | 356 | 584 | 1.111 | 1.101 |
| GeoLDM | <u>2.37</u> | **587** | **340** | <u>522</u> | <u>1.108</u> | <u>1.025</u> |
| GCDM | **1.97** | <u>602</u> | <u>344</u> | **479** | **0.844** | **0.689** |
| QM9 (Lower-bound) | 0.10 | 64 | 39 | 36 | 0.043 | 0.040 |

| Task | $\alpha\downarrow$ | $\Delta\epsilon\downarrow$ | $\epsilon_{HOMO}\downarrow$ | $\epsilon_{LUMO}\downarrow$ | $\mu\downarrow$ | $C_v\downarrow$ |
|---|---|---|---|---|---|---|
| Units | Bohr$^3$ | meV | meV | meV | D | $\frac{cal}{mol}$ K |
| GeoLDM | $2.77 \pm 0.12$ | <u>$655 \pm 20.57$</u> | $357 \pm 5.68$ | <u>$565 \pm 10.62$</u> | <u>$1.089 \pm 0.02$</u> | $1.070 \pm 0.04$ |
| GCDM | **$1.99 \pm 0.01$** | **$595 \pm 14.34$** | **$346 \pm 1.23$** | **$480 \pm 6.58$** | **$0.855 \pm 0.00$** | **$0.698 \pm 0.01$** |

| Metric | $\alpha$ PB-Valid (%) ↑ | $\Delta\epsilon$ PB-Valid (%) ↑ | $\epsilon_{HOMO}$ PB-Valid (%) ↑ | $\epsilon_{LUMO}$ PB-Valid (%) ↑ | $\mu$ PB-Valid (%) ↑ | $C_v$ PB-Valid (%) ↑ |
|---|---|---|---|---|---|---|
| GeoLDM | $93.7 \pm 0.5$ | $92.8 \pm 0.3$ | $93.9 \pm 0.4$ | $93.3 \pm 0.6$ | $93.2 \pm 1.3$ | $92.5 \pm 0.8$ |
| GCDM | $92.3 \pm 0.3$ | $92.5 \pm 0.8$ | $92.7 \pm 0.5$ | $92.7 \pm 0.6$ | $92.4 \pm 0.4$ | $91.7 \pm 0.4$ |

The results in the top half of the table are reported in terms of the MAE for molecular property prediction by an EGNN classifier $\phi_c$ on a QM9 subset, with results listed for GCDM-generated samples as well as for four separate baseline methods. The results in the bottom half of the table (where GeoLDM is retrained using its official code repository due to the unavailability of its conditional model checkpoints) are likewise listed for selected methods yet instead report (across an ensemble of three separately-trained EGNN property classifier models, each with a distinct random seed) Student's $t$-distribution 95% confidence error intervals for each property metric as well as the percentage of PoseBusters-validated (PB-Valid) de novo generated molecules. The top-1 (best) conditioning results for this task are in bold, and the second-best results are underlined.

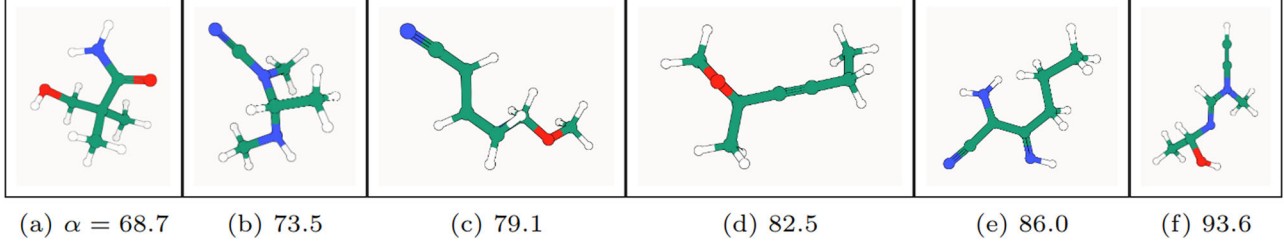

(a) $\alpha = 68.7$ (b) 73.5 (c) 79.1 (d) 82.5 (e) 86.0 (f) 93.6

**Fig. 3 | PB-valid 3D molecules generated by GCDM using increasing values of $\alpha$.** The structural characteristics of the generated molecules are gradually altered as $\alpha$ ranges from 68.7 (**a**) to 93.6 (**f**).

bound)" where a molecular property classifier $\phi_c$ predicts molecular properties given a method's generated 3D molecules and shuffled (i.e., random) property labels; and "# Atoms" where one uses the numbers of atoms in a method's generated 3D molecules to predict their molecular properties. For each baseline method, we report its mean absolute error (MAE) in terms of molecular property prediction by an ensemble of three EGNN classifiers $\phi_c$[36] as reported in ref. [25]. For GCDM, we train each conditional model by conditioning it on one of six distinct molecular property feature inputs—$\alpha$, gap, homo, lumo, $\mu$, and $C_v$—for approximately 1500 epochs using the QM9 validation split of ref. [25] as the model's training dataset and the QM9 training split of ref. [25] as the corresponding EGNN classifier ensemble's training dataset. Consequently, one can expect the gap between a method's performance and that of "QM9 (Lower-bound)" to decrease as the method more accurately generates property-specific molecules.

**Results.** We see in Table 2 that GCDM achieves the best overall results compared to all baseline methods in conditioning on a given molecular property, with conditionally-generated samples shown in Fig. 3 (Note: PSI4-computed property values[37] for (a) and (f) are 69.1 Bohr$^3$ (energy: −402 a.u.) and 89.7 Bohr$^3$ (energy: −419 a.u.), respectively, at the DFT/ B3LYP/6-31G(2DF,P) level of theory[24,38]). In particular, as shown in the bottom half of this table, GCDM surpasses the MAE results of the SOTA GeoLDM method (by 19% on average) for all six molecular properties—

$\alpha$, gap, homo, lumo, $\mu$, and $C_v$—by 28%, 9%, 3%, 15%, 21%, and 35%, respectively, while nearly matching the PB-Valid rates of GeoLDM (similar to the results in Table 1). These results qualitatively and quantitatively demonstrate that, using geometry-complete diffusion, GCDM enables notably precise generation of 3D molecules with specific molecular properties (e.g., $\alpha$—polarizability).

**Unconditional 3D molecule generation—GEOM-Drugs**
The second dataset used in our experiments, the GEOM-Drugs dataset, is a well-known source of large, 3D molecular conformers for downstream machine learning tasks. It contains 430k molecules, each with 44 atoms on average and with up to as many as 181 atoms after hydrogen atoms are imputed for each molecule following dataset postprocessing as in ref. [25]. For this experiment, we collect the 30 lowest-energy conformers corresponding to a molecule and task each baseline method with generating new molecules with 3D positions and types for each constituent atom. Here, we also adopt the negative log-likelihood, atom stability, and molecule stability metrics as defined in the "Unconditional 3D molecule generation—QM9" section and train GCDM using the same hyperparameters as listed in Supplementary Note B.2, with the exception of training for approximately 75 epochs on GEOM-Drugs.

**Baselines.** In this experiment, we compare GCDM to several state-of-the-art baseline methods for 3D molecule generation on GEOM-Drugs. Similar to our experiments on QM9, in addition to including a reference

## Table 3 | Comparison of GCDM with baseline methods for 3D molecule generation

| Type | Method | NLL ↓ | AS (%) ↑ | MS (%) ↑ |
|---|---|---|---|---|
| NF | E-NF | – | 75.0 | 0.0 |
| DDPM | GDM | −14.2 | 75.0 | 0.0 |
| | GDM-aug | −58.3 | 77.7 | 0.0 |
| | EDM | −137.1 | 81.3 | 0.0 |
| | Bridge | – | 81.0 ± 0.7 | 0.0 |
| | Bridge + Force | – | 82.4 ± 0.8 | 0.0 |
| LDM | GraphLDM | – | 76.2 | 0.0 |
| | GraphLDM-aug | – | 79.6 | 0.0 |
| | GeoLDM | – | 84.4 | 0.0 |
| GC-DDPM—Ours | GCDM w/o Frames | 769.7 | 88.0 ± 0.3 | 3.4 ± 0.3 |
| | GCDM w/o SMA | 3505.5 | 43.9 ± 3.6 | 0.1 ± 0.0 |
| | GCDM | **−234.3** | **89.0 ± 0.8** | **5.2 ± 1.1** |
| Data | | | 86.5 | 2.8 |

| Method | NLL ↓ | AS (%) ↑ | MS (%) ↑ | Val (%) ↑ | Val and Uniq (%) ↑ | Novel (%) ↑ | PB-Valid (%) ↑ |
|---|---|---|---|---|---|---|---|
| GeoLDM | – | 84.4 ± 0.1 | 0.6 ± 0.1 | **99.5 ± 0.1** | **99.4 ± 0.1** | – | 38.3 ± 0.5 |
| GCDM | **−215.1 ± 3.8** | **88.1 ± 0.1** | **4.3 ± 0.4** | 95.5 ± 0.1 | 95.5 ± 0.1 | 95.5 ± 0.1 | **77.0 ± 0.1** |

The results in the top half of the table are reported in terms of each method's negative log-likelihood, atom stability, and molecule stability with standard deviations (±) across three runs on GEOM-Drugs, each drawing 10,000 samples from the model. The results in the bottom half of the table are for methods specifically evaluated across five runs on QM9 using Student's *t*-distribution 95% confidence intervals for per-metric errors, additionally with validity and uniqueness (Val and Uniq), novelty (Novel), and PoseBusters validity (PB-Valid) defined likewise as in the "Unconditional 3D molecule generation —QM9" section; The top-1 (best) results for this task are in bold, and the second-best results are underlined.

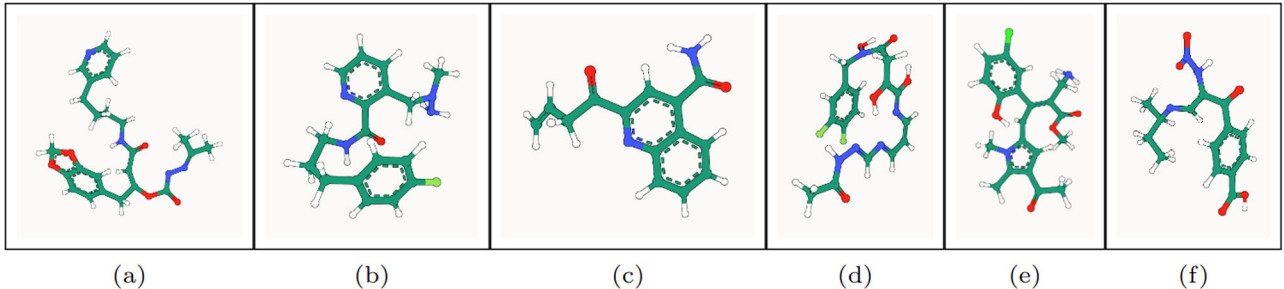

(a)  (b)  (c)  (d)  (e)  (f)

**Fig. 4 | PB-valid 3D molecules generated by GCDM for the GEOM-Drugs dataset.** The corresponding SMILES strings for these generated large molecules, from left to right, are as follows: **a** CC(C)=N[N]C(=O)O[C]([CH]C(=O)NCCCCc1cccnc1) Cc1ccc2c(c1)OCO2, **b** CN(N)Cc1cccnc1C(=O)NCCCc1ccc(F)cc1, **c** C=CCC(=O) c1cc(C(N)=O)c2ccccc2n1, **d** CC(=O)N/N=C/N=C/C=C\N=C(/O)[C](O)CC(=O) N(O)Cc1ccc(F)c(F)c1, **e** COC(=O)/C(CN)=C(\[CH]c1cc(C(C)=O)c(C)n1C) c1cc(Cl)ccc1O, and **f** CC[C@@H](C)/N=C/[C](N[N+](=O)[O-])C(=O) c1ccc(C(=O)O)cc1.

point for molecule quality metrics using GEOM-Drugs itself (i.e., Data), here we also compare against E-NF, GDM, GDM-aug, EDM, Bridge along with its variant Bridge + Force, as well as GraphLDM, GraphLDM-aug, and GeoLDM. As in the "Unconditional 3D molecule generation— QM9" section, each method's results in the top half (bottom half) of the table are reported as the mean and standard deviation (mean and Student's *t*-distribution 95% confidence interval) (±) of each metric across three (five) test runs on GEOM-Drugs.

**Results.** To start, Table 3 displays an interesting phenomenon that is important to note: due to the size and atomic complexity of GEOM-Drugs' molecules and the subsequent errors accumulated when estimating bond types based on such inter-atom distances, the baseline results for the molecule stability metrics measured here (i.e., Data) are much lower than those collected for the QM9 dataset. Thus, reporting additional chemical and structural validity metrics (e.g., PB-Valid) for comparison is crucial to accurately assess a method's performance in this context, which we do in the bottom half of Table 3. Nonetheless, for GEOM-Drugs, GCDM supersedes EDM's SOTA negative log-likelihood results by 57% and advances GeoLDM's SOTA atom and molecule

stability results by 4% and more than sixfold, respectively. More importantly, however, GCDM can generate a significant proportion of PB-valid large molecules, surpassing even the reference molecule stability rate of the GEOM-Drugs dataset (i.e., 2.8) by 54%, demonstrating that geometric diffusion models such as GCDM can not only effectively generate valid large molecules but can also generalize beyond the native distribution of stable molecules within GEOM-Drugs.

Figure 4 illustrates PoseBusters-valid examples of large molecules generated by GCDM at the scale of GEOM-Drugs. As an example of the notion that GCDM produces low energy structures for a generated molecular graph, the free energies for Fig. 4a, f were computed to be −3 kcal/mol and −2 kcal/mol, respectively, using CREST 2.12[39] at the GFN2-XTB level of theory (which matches the corresponding free energy distribution mean for the GEOM-Drugs dataset (−2.5 kcal/mol) as illustrated in Fig. 2 of ref. 40). Lastly, to detect whether a method, in aggregate, generates molecules with unlikely 3D conformations, a generated molecule's energy ratio is defined as in ref. 23 to be the ratio of the molecule's UFF-computed energy[41] and the mean of 50 RDKit ETKDGv3-generated conformers[42] of the same molecular graph. Note that, as discussed by ref. 43, generated molecules with an energy ratio greater than 7 are considered to have highly unlikely 3D

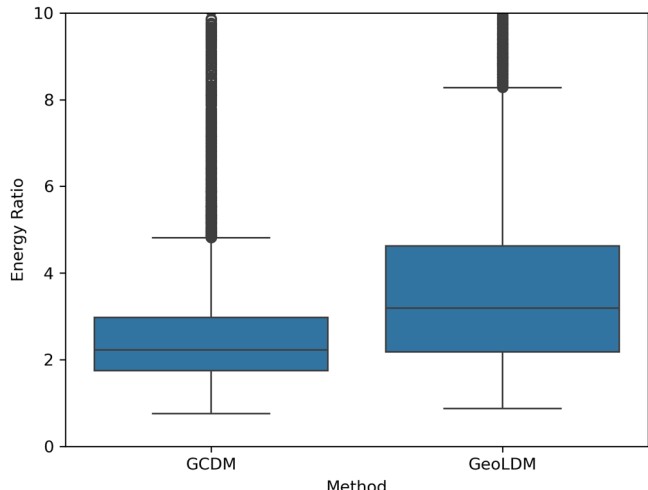

**Fig. 5 | A comparison of the energy ratios[23] of 10,000 large 3D molecules generated by GCDM and GeoLDM, a baseline state-of-the-art method.** Employing Student's $t$-distribution 95% confidence intervals, GCDM achieves a mean energy ratio of 2.98 ± 0.13, whereas GeoLDM yields a mean energy ratio of 4.19 ± 0.09.

conformations. Subsequently, Fig. 5 reveals that the average energy ratio of GCDM's large 3D molecules is notably lower and more tightly bounded compared to GeoLDM, the baseline SOTA method for this task, indicating that GCDM also generates more energetically-stable 3D molecule conformations compared to prior methods.

### Property-guided 3D molecule optimization—QM9

To evaluate whether molecular diffusion models can not only generate new 3D molecules but can also optimize existing small molecules using molecular property guidance, we adopt the QM9 dataset for the following experiment. First, we use an unconditional GCDM model to generate 1000 3D molecules using 10 time steps of time-scaled reverse diffusion (to leave such molecules in an unoptimized state), and then we provide these molecules to a separate property-conditional diffusion model for optimization of the molecules towards the conditional model's respective property. This conditional model accepts these 3D molecules as intermediate states for 100 and 250 time steps of property-guided optimization of the molecules' atom types and 3D coordinates. Lastly, we repurpose our experimental setup from the "Property-conditional 3D molecule generation—QM9" section to score these optimized molecules using an ensemble of external property classifier models to evaluate (1) how much the optimized molecules' predicted property values have been improved for the respective property (first metric) and (2) whether and how much the optimized molecules' stability (as defined in the "Unconditional 3D molecule generation—QM9" section) has been changed during optimization (second metric).

**Baselines.** Baseline methods for this experiment include EDM[25] and GCDM, where both methods use similar experimental setups for evaluation. Our baseline methods also include property-specificity and molecule stability measures of the initial (unconditional) 3D molecules to demonstrate how much molecular diffusion models can modify or improve these existing 3D molecules in terms of how property-specific and stable they are. As in the "Property-conditional 3D molecule generation—QM9" section, property specificity is measured in terms of the corresponding property classifier's MAE for a given molecule with a targeted property value, reporting the mean and Student's $t$-distribution 95% confidence interval for each property MAE across an ensemble of three corresponding classifiers. Molecular stability (i.e., Mol Stable (%)), here abbreviated at *MS*, is defined as in the "Unconditional 3D molecule generation—QM9" section.

**Results.** In this section, we quantitatively explore (in Fig. 6) whether and how much generative models can reduce the property-specific MAE and improve the molecular stability of a batch of existing 3D molecules. In particular, Fig. 6 showcases a practical finding: geometric diffusion models such as GCDM can effectively be repurposed as 3D molecule optimization methods with minimal modifications, improving both a molecule's stability and property specificity. This finding empirically supports the idea that molecular denoising diffusion models may be applied in the optimization stage of the typical drug discovery pipeline[44] to experiment with a wider range of potential drug candidates (post-optimization) more quickly than previously possible. Simultaneously, the baseline EDM method fails to consistently optimize the stability and property specificity of existing 3D molecules, which suggests that geometric methods such as GCDM are theoretically and empirically better suited for such tasks. Notably, on average, with 100 time steps GCDM improves the stability of the initial molecules by over 25% and their specificity for each molecular property by over 27%, whereas for the properties it can optimize with 100 time steps, EDM improves the stability of the molecules by 13% and their property specificity by 15%. Lastly, it is worth noting that increasing the number of optimization time steps from 100 to 250 steps inconsistently leads to further improvements to molecules' stability and property specificity, indicating that the optimization trajectory likely reaches a local minimum around 100 time steps and hence rationalizes reducing the required compute time for optimizing 1000 molecules e.g., from 15 min (for 250 steps) to 5 min (for 100 steps).

### Protein-conditional 3D molecule generation

To investigate whether geometry-complete methods can enhance the ability of molecular diffusion models to generate 3D models within a given protein pocket (i.e., to perform structure-based drug design (SBDD)), in this experiment, we adopt the standard Binding MOAD[45] and CrossDocked[46] datasets for training and evaluation of GCDM-SBDD, our geometry-complete, diffusion generative model based on GCPNET++ that extends the diffusion framework of ref. 47 for protein pocket-aware molecule generation. The Binding MOAD dataset consists of 100,000 high-quality protein-ligand complexes for training and 130 proteins for testing, with a 30% sequence identity threshold being used to define this cross-validation split. Similarly, the CrossDocked dataset contains 40,484 high-quality protein-ligand complexes split between training (40,354) and test (100) partitions using proteins' enzyme commission numbers as described by ref. 47.

**Baselines.** Baseline methods for this experiment include DiffSBDD-cond[47] and DiffSBDD-joint[47]. We compare these methods to our proposed geometry-complete protein-aware diffusion model, GCDM-SBDD, using metrics that assess the properties, and thereby the quality, of each method's generated molecules. These molecule-averaged metrics include a method's average Vina score (computed using QuickVina 2.1)[48] as a physics-based estimate of a ligand's estimated binding affinity with a target protein, measured in units of kcal/mol (lower is better); average drug likeliness QED[49] (computed using RDKit 2022.03.2); average synthesizability[50] (computed using the procedure introduced by ref. 51) as an increasing measure of the ease of synthesizing a given molecule (higher is better); on average how many rules of Lipinski's rule of five are satisfied by a ligand[52] (computed compositionally using RDKit 2022.03.2); and average diversity in mean pairwise Tanimoto distances[53,54] (derived manually using fingerprints and Tanimoto similarities computed by RDKit 2022.03.2). Following established conventions for 3D molecule generation[25], the size of each ligand to generate was determined using the ligand size distribution of the respective training dataset. Note that, in this context, "joint" and "cond" configurations represent generating a molecule for a protein target, respectively, with and without also modifying the coordinates of the binding pocket within the protein target. Also note that, similar to our experiments in the "Unconditional 3D molecule generation—QM9", "Property-conditional 3D molecule generation—QM9", "Unconditional 3D molecule generation—GEOM-Drugs" and "Property-guided 3D molecule optimization

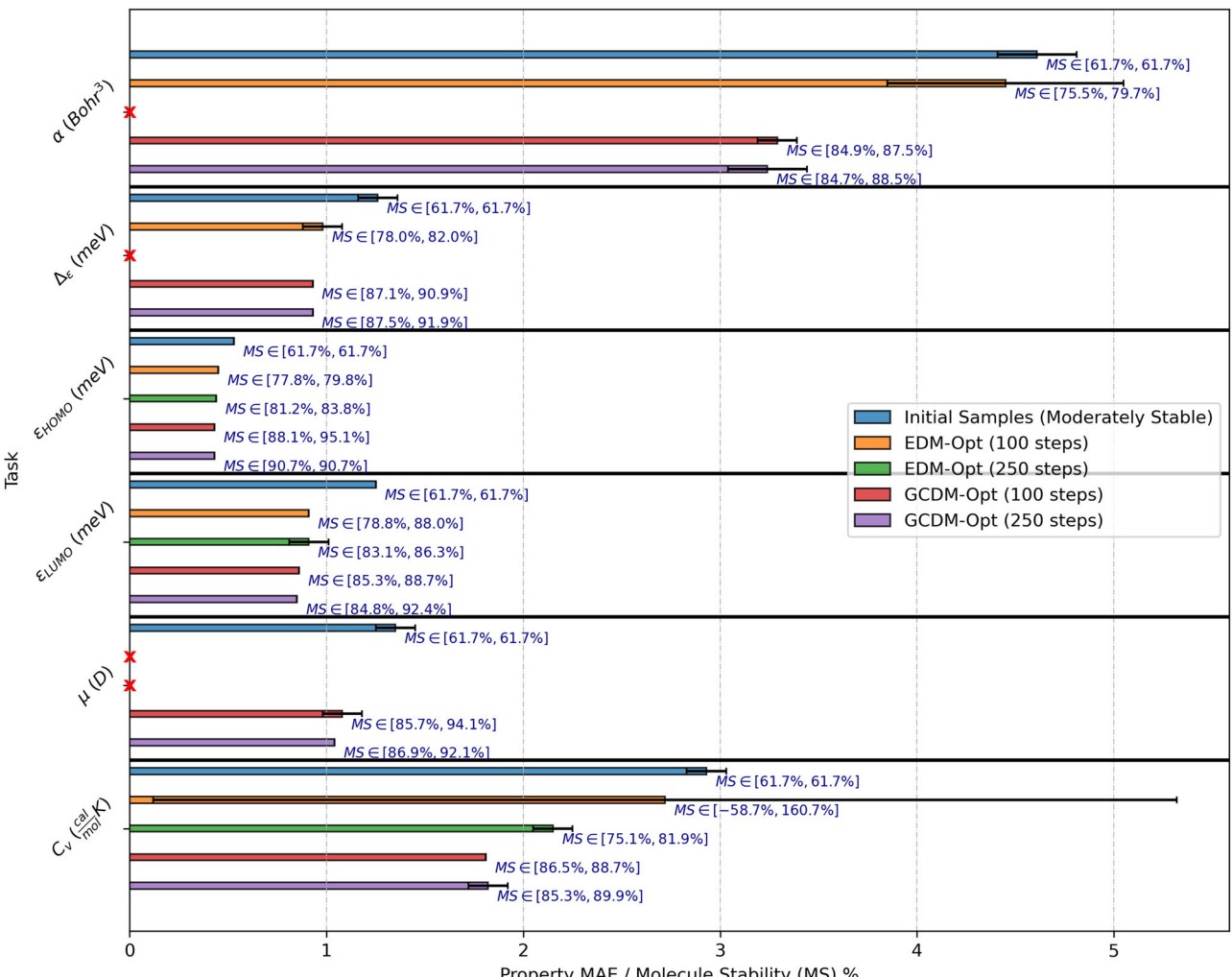

**Fig. 6 | Comparison of GCDM with baseline methods for property-guided 3D molecule optimization.** The results are reported in terms of molecular stability ($MS$) and the MAE for molecular property prediction by an ensemble of three EGNN classifiers $\phi_c$ (each trained on the same QM9 subset using a distinct random seed) yielding corresponding Student's $t$-distribution 95% confidence intervals, with results listed for EDM and GCDM-optimized samples as well as the molecule generation baseline ("Initial Samples"). Note that $x$ denotes a missing bar representing outlier property MAEs greater than 50. Alternatively, tabular results are given in Table 1 of Supplementary Results C.1.

—QM9" sections, the GCDM-SBDD model uses 9 GCP message-passing layers along with 256 (64) and 32 (16) invariant (equivariant) node and edge features, respectively.

**Results**. Table 4 shows that, across both of the standard SBDD datasets (i.e., Binding MOAD and CrossDocked), GCDM-SBDD generates more clash-free (PB-Valid) and lower energy (Vina) molecules compared to prior methods. Moreover, across all other metrics, GCDM-SBDD achieves comparable or better results in terms of drug-likeness measures (e.g., QED) and comparable results for all other molecule metrics without performing any hyperparameter tuning due to compute constraints. These results suggest that GCDM, with GCPNET++ as its denoising neural network, not only works well for de novo 3D molecule generation but also protein target-specific 3D molecule generation, notably expanding the number of real-world application areas of GCDM. Concretely, GCDM-SBDD improves upon DiffSBDD's average Vina energy scores by 8% on average across both datasets while generating more than twice as many PB-valid "candidate" molecules for the more challenging Binding MOAD dataset.

As suggested by ref. 23, the gap between the PB-Valid ratios in Table 4 without and with protein-ligand steric clashes considered for both GCDM-SBDD and DiffSBDD suggests that deep learning-based drug design methods for targeted protein pockets can likely benefit significantly from interaction-aware molecular dynamics relaxation following protein-conditional molecule generation, which may allow for many generated "candidate" molecules to have their PB validity "recovered" by such relaxation. Nonetheless, Fig. 7 demonstrates that GCDM can consistently generate clash-free realistic and diverse 3D molecules with low Vina energies for unseen protein targets.

## Conclusions

While previous methods for 3D molecule generation have possessed insufficient geometric and molecular priors for scaling well to a variety of molecular datasets, in this work, we introduced a geometry-complete diffusion model (GCDM) that establishes a clear performance advantage over previous methods, generating more realistic, stable, valid, unique, and property-specific 3D molecules, while enabling the generation of many large 3D molecules that are energetically stable as well as chemically and structurally valid. Moreover, GCDM does so without complex modeling techniques such as latent diffusion, which suggests that GCDM's results could likely be further improved by expanding upon these techniques[33]. Although GCDM's results here are promising, since it (like previous methods) requires fully-connected graph attention as well as 1000 time steps to generate a high-quality batch of 3D molecules, using it to generate several

**Table 4 | Evaluation of generated molecules for target protein pockets from the Binding MOAD (BM) and CrossDocked (CD) test datasets**

| Dataset | Method | Vina (kcal/mol, ↓) | QED (↑) | SA (↑) | Lipinski (↑) | Diversity (↑) | PB-Valid (%) (↑) |
|---|---|---|---|---|---|---|---|
| BM | DiffSBDD-cond ($C\alpha$) | −5.784 ± 0.03 | 0.433 ± 0.00 | 0.616 ± 0.00 | 4.719 ± 0.01 | 0.848 ± 0.00 | 16.6 ± 0.6/1.7 ± 0.2 |
| | DiffSBDD-joint ($C\alpha$) | −5.882 ± 0.05 | 0.474 ± 0.00 | 0.631 ± 0.00 | 4.835 ± 0.01 | 0.852 ± 0.00 | 10.7 ± 0.5/0.7 ± 0.1 |
| | GCDM-SBDD-cond ($C\alpha$) (Ours) | −**6.250** ± 0.03 | <u>0.465</u> ± 0.00 | <u>0.618</u> ± 0.00 | 4.661 ± 0.01 | 0.806 ± 0.00 | **40.8** ± 0.8/**6.8** ± 0.4 |
| | GCDM-SBDD-joint ($C\alpha$) (Ours) | <u>−6.159</u> ± 0.06 | 0.459 ± 0.00 | 0.584 ± 0.00 | 4.609 ± 0.02 | 0.794 ± 0.00 | <u>37.3</u> ± 0.8 / <u>2.0</u> ± 0.2 |
| | Reference | −8.328 ± 0.04 | 0.602 ± 0.00 | 0.336 ± 0.00 | 4.838 ± 0.01 | – | – |
| CD | DiffSBDD-cond ($C\alpha$) | −5.540 ± 0.03 | 0.449 ± 0.00 | 0.636 ± 0.00 | 4.735 ± 0.01 | 0.818 ± 0.00 | 40.7 ± 1.0/12.4 ± 0.6 |
| | DiffSBDD-joint ($C\alpha$) | −5.735 ± 0.05 | 0.420 ± 0.00 | 0.662 ± 0.00 | 4.859 ± 0.01 | 0.890 ± 0.00 | 34.1 ± 0.9/6.2 ± 0.5 |
| | GCDM-SBDD-cond ($C\alpha$) (Ours) | −**5.955** ± 0.04 | <u>0.457</u> ± 0.00 | <u>0.640</u> ± 0.00 | <u>4.758</u> ± 0.02 | 0.795 ± 0.00 | 38.1 ± 1.0/**15.7** ± 0.7 |
| | GCDM-SBDD-joint ($C\alpha$) (Ours) | <u>−5.870</u> ± 0.03 | **0.458** ± 0.00 | 0.631 ± 0.00 | 4.701 ± 0.02 | 0.810 ± 0.00 | **46.8** ± 1.0/6.5 ± 0.5 |
| | Reference | −6.871 ± 0.04 | 0.476 ± 0.00 | 0.728 ± 0.00 | 4.340 ± 0.00 | – | – |

Our proposed method, GCDM-SBDD, achieves the best results for the metrics listed in bold and the second-best results for the metrics underlined. For each metric, a method's mean and Student's *t*-distribution 95% confidence error interval (±) is reported over 100 generated molecules for each test pocket. Additionally, the PB-Valid metric is defined as the percentage of generated molecules that pass all docking-relevant structural and chemical sanity checks proposed by ref. 23, with the validity ratio to the left (right) of each/denoting the percentage of valid molecules without (with) consideration of protein-ligand steric clashes.

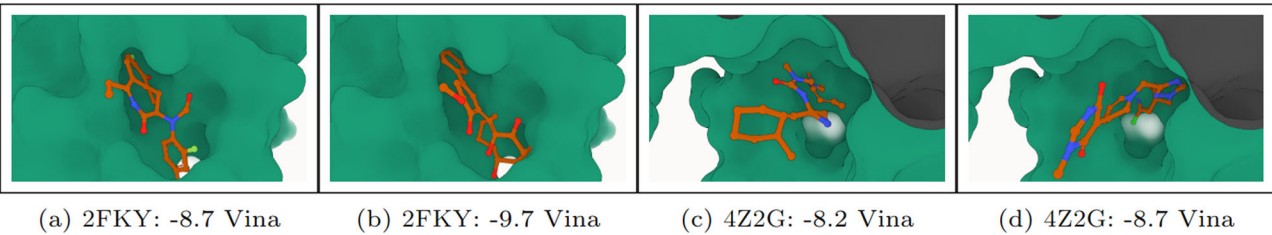

(a) 2FKY: -8.7 Vina    (b) 2FKY: -9.7 Vina    (c) 4Z2G: -8.2 Vina    (d) 4Z2G: -8.7 Vina

**Fig. 7 | Protein pocket-specific 3D molecules generated by GCDM-SBDD.** GCDM-SBDD molecules generated for BM (**a, b**) and CD (**c, d**) test proteins. Vina energy scores for these selected pocket-binding molecules range from −8.2 (**c**) to −9.7 (**b**).

thousand large molecules can take a notable amount of time (e.g., 15 minutes to generate 250 new large molecules). As such, future research with GCDM could involve adding new time-efficient graph construction or sampling algorithms[55] or exploring the impact of higher-order (e.g., type-2 tensor) yet efficient geometric expressiveness[56] on 3D generative models to accelerate sample generation and increase sample quality. Furthermore, integrating additional external tools for assessing the quality and rationality of generated molecules[57] is a promising direction for future work.

## Methods
### Problem setting
In this work, our goal is to generate new 3D molecules either unconditionally or conditioned on user-specified properties. We represent a molecular point cloud (e.g., 3D molecule) as a fully-connected 3D graph $\mathcal{G} = (\mathcal{V}, \mathcal{E})$ with $\mathcal{V}$ and $\mathcal{E}$ representing the graph's sets of nodes and edges, respectively, and $N = |\mathcal{V}|$ and $E = |\mathcal{E}|$ representing the numbers of nodes and edges in the graph, accordingly. In addition, $\mathbf{X} = (\mathbf{x}_1, \mathbf{x}_2, ..., \mathbf{x}_N) \in \mathbb{R}^{N \times 3}$ represents the respective Cartesian coordinates for each node (i.e., atom). Each node in $\mathcal{G}$ is described by scalar features $\mathbf{H} \in \mathbb{R}^{N \times h}$ and $m$ vector-valued features $\chi \in \mathbb{R}^{N \times (m \times 3)}$. Likewise, each edge in $\mathcal{G}$ is described by scalar features $\mathbf{E} \in \mathbb{R}^{E \times e}$ and $x$ vector-valued features $\xi \in \mathbb{R}^{E \times (x \times 3)}$. Then, let $\mathcal{M} = [\mathbf{X}, \mathbf{H}]$ represent the molecules (i.e., atom coordinates and atom types) our method is tasked with generating, where $[\cdot, \cdot]$ denotes the concatenation of two variables. Important to note is that the input features $\mathbf{H}$ and $\mathbf{E}$ are invariant to 3D roto-translations, whereas the input vector features $\mathbf{X}, \chi$ and $\xi$ are equivariant to 3D roto-translations. Lastly, in particular, we design a denoising neural network $\boldsymbol{\Phi}$ to be equivariant to 3D roto-translations (i.e., SE(3)-equivariant) by defining it such that its internal operations and outputs match corresponding 3D roto-translations acting upon its inputs.

### Overview of GCDM
We will now introduce GCDM, a new Geometry-Complete SE(3)-Equivariant Diffusion Model. GCDM defines a joint noising process on equivariant atom coordinates $\mathbf{x}$ and invariant atom types $\mathbf{h}$ to produce a noisy representation $\mathbf{z} = [\mathbf{z}^{(\mathbf{x})}, \mathbf{z}^{(\mathbf{h})}]$ and then learns a generative denoising process using the newly-proposed GCPNET++ model (see Supplementary Methods A.1), which desirably contains two distinct feature channels for scalar and vector features, respectively, and supports geometry-complete and chirality-aware message-passing[58].

As an extension of the DDPM framework[59] outlined in Supplementary Methods A.2.1, GCDM is designed to generate molecules in 3D while maintaining SE(3) equivariance, in contrast to previous methods that generate molecules solely in 1D[60], 2D[61], or 3D modalities without considering chirality[9,25]. GCDM generates molecules by directly placing atoms in continuous 3D space and assigning them discrete types, which is accomplished by modeling forward and reverse diffusion processes, respectively:

$$\underbrace{q(\mathbf{z}_{1:T}|\mathbf{z}_0) = \prod_{t=1}^{T} q(\mathbf{z}_t|\mathbf{z}_{t-1})}_{\text{Forward}} \qquad \underbrace{p_{\boldsymbol{\Phi}}(\mathbf{z}_{0:T-1}|\mathbf{z}_T) = \prod_{t=1}^{T} p_{\boldsymbol{\Phi}}(\mathbf{z}_{t-1}|\mathbf{z}_t)}_{\text{Reverse}}$$

Overall, these processes describe a latent variable model $p_{\boldsymbol{\Phi}}(\mathbf{z}_0) = \int p_{\boldsymbol{\Phi}}(\mathbf{z}_{0:T}) d\mathbf{z}_{1:T}$ given a sequence of latent variables $\mathbf{z}_0, \mathbf{z}_1, ..., \mathbf{z}_T$ matching the dimensionality of the data $\mathcal{M} \sim p(\mathbf{z}_0)$. As illustrated in Fig. 1, the forward

process (directed from right to left) iteratively adds noise to an input, and the learned reverse process (directed from left to right) iteratively denoises a noisy input to generate new examples from the original data distribution. We will now proceed to formulate GCDM's joint diffusion process and its remaining practical details.

## Joint molecular diffusion

Recall that our model's molecular graph inputs, $\mathcal{G}$, associate with each node a 3D position $\mathbf{x}_i \in \mathbb{R}^3$ and a feature vector $\mathbf{h}_i \in \mathbb{R}^h$. By way of adding random noise to these model inputs at each time step $t$ via a fixed, Markov chain variance schedule $\sigma_1^2, \sigma_2^2, \ldots, \sigma_T^2$, we can define a joint molecular diffusion process for equivariant atom coordinates $\mathbf{x}$ and invariant atom types $\mathbf{h}$ as the product of two distributions[25]:

$$q(\mathbf{z}_t|\mathbf{z}_{t-1}) = \mathcal{N}_{xh}(\mathbf{z}_t|\alpha_t\mathbf{z}_{t-1}, \sigma_t^2\mathbf{I}). \tag{1}$$

where $\mathcal{N}_{xh}$ serves as concise notation to denote the product of two normal distributions; the first distribution, $\mathcal{N}_x$, represents the noised node coordinates; the second distribution, $\mathcal{N}_h$, represents the noised node features; and $\alpha_t = \sqrt{1 - \sigma_t^2}$ following the variance preserving process of ref. 59. With $\alpha_{t|s} = \alpha_t/\alpha_s$ and $\sigma_{t|s}^2 = \sigma_t^2 - \alpha_{t|s}^2\sigma_s^2$ for any $t > s$, we can directly obtain the noisy data distribution $q(\mathbf{z}_t|\mathbf{z}_0)$ at any time step $t$:

$$q(\mathbf{z}_t|\mathbf{z}_0) = \mathcal{N}_{xh}(\mathbf{z}_t|\alpha_{t|0}\mathbf{z}_0, \sigma_{t|0}^2\mathbf{I}). \tag{2}$$

Bayes Theorem then tells us that if we then define $\boldsymbol{\mu}_{t \to s}(\mathbf{z}_t, \mathbf{z}_0)$ and $\sigma_{t \to s}$ as:

$$\boldsymbol{\mu}_{t \to s}(\mathbf{z}_t, \mathbf{z}_0) = \frac{\alpha_s\sigma_{t|s}^2}{\sigma_t^2}\mathbf{z}_0 + \frac{\alpha_{t|s}\sigma_s^2}{\sigma_t^2}\mathbf{z}_t \text{ and } \sigma_{t \to s} = \frac{\sigma_{t|s}\sigma_s}{\sigma_t},$$

we have that the inverse of the noising process, the true denoising process, is given by the posterior of the transitions conditioned on $\mathcal{M} \sim \mathbf{z}_0$, a process that is also Gaussian[25]:

$$q(\mathbf{z}_s|\mathbf{z}_t, \mathbf{z}_0) = \mathcal{N}(\mathbf{z}_s|\boldsymbol{\mu}_{t \to s}(\mathbf{z}_t, \mathbf{z}_0), \sigma_{t \to s}^2\mathbf{I}). \tag{3}$$

## Parametrization of the reverse process

**Noise parametrization.** We now need to define the learned generative reverse process that denoises pure noise into realistic examples from the original data distribution. Towards this end, we can directly use the noise posteriors $q(\mathbf{z}_s|\mathbf{z}_t, \mathbf{z}_0)$ of Eq. A12 within Supplementary Methods A.2.1 after sampling $\mathbf{z}_0 \sim (\mathcal{M} = [\mathbf{x}, \mathbf{h}])$. However, to do so, we must replace the input variables $\mathbf{x}$ and $\mathbf{h}$ with the approximations $\hat{\mathbf{x}}$ and $\hat{\mathbf{h}}$ predicted by the denoising neural network $\Phi$:

$$p_\Phi(\mathbf{z}_s|\mathbf{z}_t) = \mathcal{N}_{xh}(\mathbf{z}_s|\boldsymbol{\mu}_{\Phi_{t \to s}}(\mathbf{z}_t, \tilde{\mathbf{z}}_0), \sigma_{t \to s}^2\mathbf{I}), \tag{4}$$

where the values for $\tilde{\mathbf{z}}_0 = [\hat{\mathbf{x}}, \hat{\mathbf{h}}]$ depend on $\mathbf{z}_t$, $t$, and the denoising neural network $\Phi$. GCDM then parametrizes $\boldsymbol{\mu}_{\Phi_{t \to s}}(\mathbf{z}_t, \tilde{\mathbf{z}}_0)$ to predict the noise $\hat{\boldsymbol{\epsilon}} = [\hat{\boldsymbol{\epsilon}}^{(x)}, \hat{\boldsymbol{\epsilon}}^{(h)}]$, which represents the noise individually added to $\hat{\mathbf{x}}$ and $\hat{\mathbf{h}}$. We can then use the predicted $\hat{\boldsymbol{\epsilon}}$ to derive:

$$\tilde{\mathbf{z}}_0 = [\hat{\mathbf{x}}, \hat{\mathbf{h}}] = \mathbf{z}_t/\alpha_t - \hat{\boldsymbol{\epsilon}} \cdot \sigma_t/\alpha_t. \tag{5}$$

**Invariant likelihood.** Ideally, we desire for a 3D molecular diffusion model to assign the same likelihood to a generated molecule even after arbitrarily rotating or translating it in 3D space. To ensure the model achieves this desirable property for $p_\Phi(\mathbf{z}_0)$, we can leverage the insight that an invariant distribution composed of an equivariant transition function yields an invariant distribution[9,25,27]. Moreover, to address the translation invariance issue raised by ref. 27 in the context of handling a distribution over 3D coordinates, we adopt the zero center of gravity trick proposed by ref. 9 to define $\mathcal{N}_x$ as a normal distribution on the subspace defined by $\sum_i \mathbf{x}_i = \mathbf{0}$. In contrast, to handle node features $\mathbf{h}_i$ that are

invariant to roto-translations, we can instead use a conventional normal distribution $\mathcal{N}$. As such, if we parametrize the transition function $p_\Phi$ using an SE(3)-equivariant neural network after using the zero center of gravity trick of ref. 9, the model will have achieved the desired likelihood invariance property.

## Geometry-complete denoising network

Crucially, to satisfy the desired likelihood invariance property described in the "Parametrization of the reverse process" section while optimizing for model expressivity and runtime, GCDM parametrizes the denoising neural network $\Phi$ using GCPNET++, an enhanced version of the SE(3)-equivariant GCPNET algorithm[58], that we propose in Supplementary Methods A.1.2. Notably, GCPNET++ learns both scalar (invariant) and vector (equivariant) node and edge features through a chirality-sensitive graph message passing procedure, which enables GCDM to denoise its noisy molecular graph inputs using not only noisy scalar features but also noisy vector features that are derived directly from the noisy node coordinates $\mathbf{z}^{(x)}$ (i.e., $\psi(\mathbf{z}^{(x)})$). We empirically find that incorporating such noisy vectors considerably increases GCDM's representation capacity for 3D graph denoising.

## Optimization objective

Following previous works on diffusion models[25,32,59], the noise parametrization chosen for GCDM yields the following model training objective:

$$\mathcal{L}_t = \mathbb{E}_{\boldsymbol{\epsilon}_t \sim \mathcal{N}_{xh}(0,1)}\left[\frac{1}{2}w(t)\|\boldsymbol{\epsilon}_t - \hat{\boldsymbol{\epsilon}}_t\|^2\right], \tag{6}$$

where $\hat{\boldsymbol{\epsilon}}_t$ is the denoising network's noise prediction for atom types and coordinates as described above and where we empirically choose to set $w(t) = 1$ for the best possible generation results. Additionally, GCDM permits a negative log-likelihood computation using the same optimization terms as ref. 25, for which we refer interested readers to Supplementary Methods A.2.2–A.2.4.

## Data availability

The data required to train new GCDM models or reproduce our results are available under a Creative Commons Attribution 4.0 International Public License at https://zenodo.org/record/7881981[62]. Additionally, all pretrained model checkpoints are available under a Creative Commons Attribution 4.0 International Public License at https://zenodo.org/record/10995319[63].

## Code availability

The source code for GCDM is available at https://github.com/BioinfoMachineLearning/Bio-Diffusion, and the source code for structure-based drug design experiments with GCDM is separately available at https://github.com/BioinfoMachineLearning/GCDM-SBDD.

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

## Acknowledgements

The authors would like to thank Chaitanya Joshi and Roland Oruche for helpful discussions and feedback on early versions of this manuscript. In addition, the authors acknowledge that this work is partially supported by three NSF grants (DBI2308699, DBI1759934, and IIS1763246), two NIH grants (R01GM093123 and R01GM146340), three DOE grants (DE-AR0001213, DE-SC0020400, and DE-SC0021303), and the computing allocation on the Summit compute cluster provided by the Oak Ridge Leadership Computing Facility under Contract DE-AC05- 00OR22725.

## Author contributions

A.M. and J.C. conceived the project. A.M. designed the experiments. A.M. performed the experiments and collected the data. A.M. analyzed the data. J.C. secured the funding for this project. A.M. and J.C. wrote the manuscript. A.M. and J.C. edited the manuscript.

## Competing interests

The authors declare no competing interests.
