## [Peer Review File · Communications Chemistry]

This manuscript has been previously reviewed at another Nature Portfolio journal. This document only contains reviewer comments and rebuttal letters for versions considered at *Communications Chemistry*.REVIEWERS' COMMENTS:

Reviewer #1 (Remarks to the Author):

The submitted manuscript by Morehead and Cheng introduces GCDM, which is an interesting equivariant model that provides some performance benefits over existing methods in the field. I believe the work is of suitable quality for Commun. Chem. and there is a portion of the readership that will appreciate the findings. I appreciate the time-investment that the Authors have clearly dedicated to preparing a mainly well-polished manuscript. I have a few minor concerns that should be addressed prior to publication. I believe these modifications can be made without cause for additional peer review.

1. The statement about approximating Boltzmann statistics should be removed. I believe it does not impact the quality of the manuscript and its inclusion would likely require many back-and-forth discussion between myself and the reviewers. This is an unnecessary delay for a "nearly ready to go" manuscript.

2. The beginning discussion in the Results section "Figure 6 showcases a practical finding:..." should be presented after some indication of what the Readers can expect to find in Figure 6. Perhaps 1 leading sentence is sufficient.

3. Although it is not necessary, and mainly a visual preference, the amount of spacing in Figure 6 is potentially too great. I think the figure could be shrunk by decreasing spacing between bars and increasing the bar width to maintain visual clarity. However, I will leave this to the Author's discretion.

4. The fractionated SMILES strings below the images in Figure 4 should be removed.

5. The statement "...were computed to be -3 kcal/mol and -2 kcal/mol, respectively, using CREST [39]..." should be clarified to be the original crest implementation, now that there is an updated CREST. If the Author's used the recently released version, that should be clarified instead.

Author Responses to Review Comments

Reviewer #1:

The submitted manuscript by Morehead and Cheng introduces GCDM, which is an interesting equivariant model that provides some performance benefits over existing methods in the field. I believe the work is of suitable quality for Commun. Chem. and there is a portion of the readership that will appreciate the findings. I appreciate the time-investment that the Authors have clearly dedicated to preparing a mainly well-polished manuscript. I have a few minor concerns that should be addressed prior to publication. I believe these modifications can be made without cause for additional peer review.

1. The statement about approximating Boltzmann statistics should be removed. I believe it does not impact the quality of the manuscript and its inclusion would likely require many back-and-forth discussion between myself and the reviewers. This is an unnecessary delay for a “nearly ready to go” manuscript.
 - **Author Response:** Thank you for your feedback. We have removed the statement about Boltzmann statistics per your recommendation.
2. The beginning discussion in the Results section “Figure 6 showcases a practical finding:...” should be presented after some indication of what the Readers can expect to find in Figure 6. Perhaps 1 leading sentence is sufficient.
 - **Author Response:** Thank you for your suggestion here. We have added a leading sentence to prime readers for what they should expect in Figure 6.
3. Although it is not necessary, and mainly a visual preference, the amount of spacing in Figure 6 is potentially too great. I think the figure could be shrunk by decreasing spacing between bars and increasing the bar width to maintain visual clarity. However, I will leave this to the Author’s discretion.
 - **Author Response:** Thank you for your feedback. We have considered this suggestion and have decided to keep Figure 6 as is.
4. The fractionated SMILES strings below the images in Figure 4 should be removed.
 - **Author Response:** Thank you for your suggestion. We have removed these fractionated SMILES strings. Instead, we now list the full SMILES strings for each generated molecule within the caption of each respective figure.
5. The statement “...were computed to be -3 kcal/mol and -2 kcal/mol, respectively, using CREST [39]...” should be clarified to be the original crest implementation, now that there is an updated CREST. If the Author’s used the recently released version, that should be clarified instead.
 - **Author Response:** Thank you for your insightful comment. We have clarified in the main text that we used CREST 2.12 in the context of this experiment.